# A Dynamic Cellular Model as an Emerging Platform to Reproduce the Complexity of Human Vascular Calcification In Vitro

**DOI:** 10.3390/ijms25137427

**Published:** 2024-07-06

**Authors:** Elisa Ceccherini, Elisa Persiani, Manuela Cabiati, Letizia Guiducci, Silvia Del Ry, Ilaria Gisone, Alessandra Falleni, Antonella Cecchettini, Federico Vozzi

**Affiliations:** 1Institute of Clinical Physiology, National Research Council, 56124 Pisa, Italy; ceccherini@ifc.cnr.it (E.C.); elisapersiani@cnr.it (E.P.); manuela.cabiati@cnr.it (M.C.); letizia.guiducci@cnr.it (L.G.); silvia.delry@cnr.it (S.D.R.); i.gisone@student.unisi.it (I.G.); antonella.cecchettini@unipi.it (A.C.); 2Department of Clinical and Experimental Medicine, University of Pisa, 56126 Pisa, Italy; alessandra.falleni@unipi.it

**Keywords:** vascular calcification, co-culture, VSMCs, ECs, dynamic in vitro models, bioreactors

## Abstract

Vascular calcification (VC) is a cardiovascular disease characterized by calcium salt deposition in vascular smooth muscle cells (VSMCs). Standard in vitro models used in VC investigations are based on VSMC monocultures under static conditions. Although these platforms are easy to use, the absence of interactions between different cell types and dynamic conditions makes these models insufficient to study key aspects of vascular pathophysiology. The present study aimed to develop a dynamic endothelial cell–VSMC co-culture that better mimics the in vivo vascular microenvironment. A double-flow bioreactor supported cellular interactions and reproduced the blood flow dynamic. VSMC calcification was stimulated with a DMEM high glucose calcification medium supplemented with 1.9 mM NaH_2_PO_4_/Na_2_HPO_4_ (1:1) for 7 days. Calcification, cell viability, inflammatory mediators, and molecular markers (SIRT-1, TGFβ1) related to VSMC differentiation were evaluated. Our dynamic model was able to reproduce VSMC calcification and inflammation and evidenced differences in the modulation of effectors involved in the VSMC calcified phenotype compared with standard monocultures, highlighting the importance of the microenvironment in controlling cell behavior. Hence, our platform represents an advanced system to investigate the pathophysiologic mechanisms underlying VC, providing information not available with the standard cell monoculture.

## 1. Introduction

Vascular calcification (VC) is characterized by calcium–phosphate complex deposition in the medial and/or intimal layer of the arteries and heart valves [1]. VC is part of the aging process and other pathological conditions, such as diabetes, hypertension, and chronic kidney disease (CKD) [2]. The vascular vessel wall comprises an endothelial cell (EC) sheet surrounded by a layered structure consisting of contractile vascular smooth muscle cells (VSMCs). Different stimuli, mainly inflammatory mediators, lipids, and increased calcium and phosphate levels, modulate multiple signaling pathways that promote VSMCs’ phenotypic switch into osteoblast-like cells, also called the calcified phenotype [3]. Several studies explored the underlying mechanisms during the above differentiation processes, identifying multiple key mediators, including transforming growth factor β1 (TGFβ1) and Sirtuin-1 (SIRT-1) [4,5,6]. Accumulating evidence also pinpointed inflammation as a primary risk factor in perturbing vascular homeostasis and triggering VSMC differentiation. Indeed, the chronic upregulation of pro-inflammatory cytokines, such as interleukin-6 (IL-6) and interleukin (IL-1β), can activate the downstream signaling pathways contributing to VSMC osteogenic transition and mineralization [7,8]. In vitro models useful for vascular pathophysiological investigations are scarce and primarily based on the static VSMC monoculture (hereafter referred to as the ‘standard monoculture’) [9,10,11,12,13]. Although these systems are easy to use and produce important results concerning cell behavior and biological characteristics, they have objective limitations due to the absence of the proper and complex microenvironment present in vascular tissues (e.g., interactions between multiple cell types, hemodynamic conditions). To overcome these limitations, new in vitro platforms capable of recapitulating the vascular district and reflecting the events occurring in the human body are needed. This study explored the potentiality of a new dynamic model in VC investigations, in which VSMCs and ECs were used as relevant cell types associated with VC. A mathematical approach was exploited to downsize the real human tissue to the in vitro setting, using a cell composition that reflects the architecture of the vessel wall [14]. To this end, this approach considers relevant parameters such as the myocardium mass, the length and the volume of the human coronary tree, the dimension of the coronary endothelium, and the smooth muscle layer. According to the mathematical model, the ratio between the ECs and the VSMCs that have to be used in the in vitro model was defined. Moreover, the intracellular calcium content, inflammation, and VSMC differentiation were investigated using immunometric and molecular biology techniques, and the results were compared with those obtained in the standard VSMC monoculture.

## 2. Results

### 2.1. VSMC and EC Viability

The VSMCs were treated for 7 days in a calcifying medium, and then viability was assessed for each cell type. As reported in Figure 1A, the VSMCs were viable in all conditions, and, as expected, less viable when treated with the calcifying medium. However, the viability reduction for the VSMCs cultured in the calcifying medium was less pronounced in the dynamic platform with a viability of about 78% compared to the control cells. In contrast, in the standard monoculture, the viability was about 47%. This phenomenon may be due to the presence of the endothelium under stimuli of flow, conditions that are not present in standard monocultures. To underline, as shown in Figure 1B, ECs were viable in both VSMC culture conditions, with no significant changes between the two treatments.

### 2.2. VSMC Calcification

Following 7 days of treatment in the calcifying medium, a spectrophotometric determination was performed to quantitatively measure the VSMC intracellular calcium amount (Figure 2). As expected, we detected higher levels of intracellular calcium in the standard monoculture than in the dynamic co-cultures, suggesting that the surrounding microenvironment has a relevant influence on the calcification mechanisms in VSMCs. To analyze intracellular calcification in detail, a transmission electron microscopy (TEM) analysis was performed (Figure 3), highlighting intracellular calcium deposits as microcalcifications, some in the cytoplasm and others inside vesicles.

### 2.3. VSMC-Released Inflammatory Mediators

Caspase-1, IL-6 and interleukin-1β (IL-1β) are pro-inflammatory cytokines that play a key role in VC driving the VSMC activation; thus, they were quantified in VSMC culture media (Figure 4). Significant increases in caspase-1 (Figure 4A) and IL-6 (Figure 4B) levels were detected in the calcified VSMCs for both the experimental settings tested. However, for both mediators, the increase was less significant in the dynamic co-culture, confirming the importance of the surrounding microenvironment in the calcification of VSMCs.

Considering the close relationship between inflammation and VSMC calcification, a correlation analysis was performed among the intracellular calcium levels and those of the pro-inflammatory mediators (Table 1). Interestingly, positive correlations were found in both of the experimental settings, although the dynamic model shows more robust correlations between calcium levels and caspase-1 (R = 0.99, *p* = 0.0155) and IL-6 (R = 1.00; *p* = 0.0002).

### 2.4. The Microenvironment Conditions Influence VSMC Differentiation

The phenotypic switch of VSMCs is a central event in vessel remodeling and contributes to cardiovascular pathologies, including VC. A plethora of cellular effectors are involved in VSMC plasticity; thus, we focused on SIRT-1 and TGFβ1, evaluating their mRNA relative expression by real-time PCR (Figure 5). In the standard monoculture, the calcifying treatment dramatically decreased the SIRT-1 levels compared to the control cells (Figure 5A). Conversely, we did not detect significant variation between the control cells and calcified VSMCs in the dynamic setting. Moreover, a comparison of the two calcifying conditions showed significantly higher levels of SIRT-1 in the dynamic model. As far as TGFβ1 was concerned, calcifying conditions modulated its expression in an opposite manner between the standard monoculture and dynamic co-culture (Figure 5B). In the standard monoculture, calcified VSMCs exhibited a dramatic decrease in TGFβ1 levels compared to the control cells; conversely, a moderate increase was detected in the dynamic model.

## 3. Discussion

VC is an active process characterized by calcium crystal deposition within the vessel wall. VSMCs play a key role in this pathologic process, acting through osteochondrogenic differentiation, extracellular vesicle release, and calcium overload [3,15]. Blood vessel walls are comprised of different cell types, primarily ECs and VSMCs, and the interaction between these cell types is fundamental to the vasculature’s function in physiological and pathological conditions, including VC [16,17]. In the vascular wall, ECs are exposed to constant stress caused by blood flow, and this mechanical force is converted into biochemical responses that regulate the underlying VSMC behavior [18]. In this regard, ECs secrete multiple vasoactive molecules (e.g., prostanoids and nitric oxide) that maintain VSMCs in the contractile state [19]. However, injury or loss in the endothelium disturbs the crosstalk between ECs and VSMCs and triggers the phenotypic switch of VSMCs, inflammation, and functional changes that characterize the pathological remodeling of the vascular wall [20]. These pieces of evidence confirm the need for in vitro models based on the dynamic co-culture systems of ECs and VSMCs to investigate the pathophysiology of vascular diseases properly. For a long time, researchers have investigated various VC-related aspects using in vitro models based on VSMC monocultures [9,10,11,12,13]. Although these models are easy to manipulate, they exhibit intrinsic limitations related to the lack of essential elements of the in vivo system, such as the crosstalk between different cell types and the presence of a complex hemodynamic environment. In the present study, we investigated the potentiality of a novel dynamic platform, based on the EC–VSMC co-culture, in which the ratio between the two cell species was determined by applying a mathematical model that considers different in vivo parameters (i.e., the length and the volume of the human coronary tree and the myocardium mass) to downsize the real human tissue to the in vitro setting. This in vitro model was used to analyze different aspects of VC (i.e., the intracellular calcium levels, inflammatory parameters, and VSMC differentiation), and compare the results with those obtained with a standard VSMC monoculture. According to published data, VSMC viability decreased under calcifying conditions [13,14] in both of the experimental settings tested; however, the viability reduction was less marked in the dynamic platform. The dimensions of calcium deposits (i.e., macro-calcification and micro-calcification) are essential information for understanding the pathophysiology of VC in more detail. Indeed, macro-calcifications confer plaque stability, whereas micro-calcifications are implicated in plaque rupture and major cardiovascular complications [21,22,23]. TEM analyses were carried out to investigate this aspect, showing micro-calcifications either in the cytoplasm or inside vesicles. The quantitative analysis of intracellular calcium highlighted an appreciable degree of VSMC calcification in both experimental settings; however, in the dynamic co-cultures, calcium values were lower than in the standard ones. These observations could be explained considering the EC–VSMC communication as an essential event in blood vessel development and function [16,17]; thus, it is reasonable to assume that the presence of healthy endothelium under flow could promote the viability of VSMCs and counteract calcification processes. Being that VSMC activation and phenotype switch is essential for VC [3,15], the mRNA levels of TGFβ1 and SIRT-1 were analyzed using real-time PCR. Scientific evidence recognized TGFβ as a factor that inhibits VSMC phenotypic change [4,24,25,26]. In agreement with these in vitro findings, Mallat and colleagues confirmed the protective role of TGF-β signaling toward the development of atherosclerotic lesions in apoE-deficient mice [27]. Moreover, Takemura and colleagues associated the downregulation of SIRT-1, which acts on RUNX2 acetylation, with the VSMC osteogenic phenotype [28]. Conversely, increased SIRT-1 levels counteract VSMC calcification by DNA repair and promoting cell survival [29,30]. According to this study, the downregulation of TGFβ1 and SIRT-1 observed in the standard monoculture could be indicative of a more pronounced calcified phenotype than in the dynamic model, in which TGFβ1 upregulation was associated with slight, nonsignificant changes in SIRT-1 levels. The differences in TGF-β modulation between the two experimental settings can be explained by considering the different complexities of the two systems. Based on the results of Heydarkhan-Hagvall and colleagues, VSMCs showed a significantly higher TGF-β expression in the static co-culture with ECs than in the standard monoculture [31]. Similar results were obtained in the dynamic EC–VSMC co-culture, in which the EC monolayer was subjected to dynamic flow [32]. These data from the literature corroborate the results obtained from our gene expression analysis, confirming that EC–VSMC crosstalk is essential in controlling the TGF-β expression and, thus, the VSMC phenotypic switch. Indeed, in our platform, the presence of a healthy endothelial layer and the dynamic flow could influence the expression of TGF-β in VSMCs, partly attenuating the transition into the calcified form. In the standard monoculture, on the contrary, the absence of crosstalk with ECs resulted in a clear deregulation of TGF-β expression, leading to a more pronounced calcified phenotype. These observations are in accordance with the calcium quantification analysis, which showed a higher degree of VSMC calcification in the standard monoculture. The vicious circle between VSMC calcification and inflammation is well established [33], identifying IL-6 and IL-1β as potent inducers of VSMC osteogenic transition [7,8]. In both cell models, calcifying conditions increased IL-6 and caspase-1, responsible for pro-IL-1β processing into IL-1β [34]. The dynamic system showed slightly lower levels of inflammatory cytokines than the standard monoculture, but a more robust degree of correlation with calcium levels (Caspase-1: R = 0.99, *p* = 0.0155; IL-6: R = 1.00, *p* = 0.0002). Altogether, these results suggest that the system complexity influences the different degrees of calcification and inflammation, as well as the modulation of effectors related to VSMC differentiation. Indeed, VSMC monocultures represent a good high-throughput option for researchers, but cannot recapitulate the in vivo environment of vascular tissues, resulting in a poor in vitro–in vivo translation. Our dynamic platform closely captures the complexity of vascular tissues, (1) employing VSMCs and ECs as relevant cell types associated with the disease; (2) using a mathematical approach to downsize the real human tissue to the in vitro setting, and a cell composition that reflects the architecture of the vessel wall; and (3) applying a flow mimicking the hemodynamic environment. Due to its versatility and cost-effectiveness, this dynamic in vitro model might help us understand the VC mechanisms and cell–cell interactions, replicating the human vascular microenvironment more appropriately than the standard monoculture. For example, dynamic parameters can be varied ad hoc (e.g., increasing or decreasing flow rate) to study cell-specific modulations in response to different hydrodynamic VC-related conditions. In addition, it is also feasible to treat ECs with calcifying medium, thus mimicking a pathological condition that involves the entire vascular tissue, highlighting additional morphological and signaling pathway modulation implicated in VC pathophysiology.

## 4. Materials and Methods

### 4.1. Cell Cultures

Human coronary artery endothelial cells (HCAECs, Lonza, Walkersville, MD, USA) and human coronary artery smooth muscle cells (HCASMCs, Lonza) were used. HCAECs (hereafter abbreviated as ECs) were cultured in an endothelial medium consisting of endothelial cell GM MW2 growth medium with Supplements (Lonza, Walkersville, MD, USA) and Penicillin/Streptomycin for a final concentration of 100 I.U./mL and 100 μg/mL, respectively. HCASMCs (hereafter abbreviated as VSMCs) were cultured in smooth muscle medium, consisting of Medium 231 with Smooth Muscle Growth Supplement (Lonza, Walkersville, MD, USA) and Penicillin/Streptomycin for a final concentration of 100 I.U./mL and 100 μg/mL, respectively. Cells were seeded and expanded at 37 °C, 5% CO_2_, and replenished with fresh media every 3–4 days until use in the device.

### 4.2. Dynamic Co-Culture Experimental Setting

As previously reported [14], ECs and VSMCs were seeded in a LiveBox2 (LB2) double-flow bioreactor (IVTech Srl, Pisa, Italy) composed of two chambers that can be in-dependently perfused and separated by a PET membrane with 45 μm diameter circular pores (Figure 6A,B). Briefly, ECs (45,000 cells) were seeded in each upper chamber of the LB2 and cultured in the endothelial medium. VSMCs (30,000 cells) were seeded in the lower chamber of the LB2 and cultured in the smooth muscle medium (control cells). Upper chambers were connected to the peristaltic pump and were subjected to a 250 µL/min flow rate for the dynamic condition (Figure 6C). To induce VSMC calcification, the smooth muscle medium was replaced with the calcifying one composed of 1.9 mM NaH_2_PO_4_/Na_2_HPO_4_ (1:1) in DMEM high glucose [35]. The LB2 was placed in an incubator at 37 °C, with 5% CO_2_ for 7 days, and the medium was replaced after 72 h. At the end of the experiments, the culture media were collected for inflammatory marker analyses; cell viability, intracellular calcium content, and VSMC differentiation were investigated using immunometric and molecular biology techniques.

### 4.3. VSMC Standard Monoculture

VSMCs (30,000 cells) were seeded in a 12-well plate and cultured in the smooth muscle medium (control cells). To induce VSMC calcification, we used the same calcifying medium used in the dynamic setting. Cells were incubated at 37 °C, 5% CO_2_ in a humidified cell culture incubator for 7 days, replacing the calcification media after 72 h. At the end of the experiments, the culture media were collected for inflammatory marker analyses; cell viability, intracellular calcium content, and VSMC differentiation were investigated using immunometric and molecular biology techniques.

### 4.4. Cell Viability

Cell viability was determined using the CellTiter-Blue^®^ Cell Viability Assay kit (Promega, Milan, Italy), as previously reported, for both dynamic co-culture [14] and VSMC monoculture [13]. A standard curve that correlates the fluorescence values of viability and defined cell number was created to extrapolate the number of viable cells.

### 4.5. Intracellular Calcium Quantification

Calcification was assessed using the Calcium Colorimetric Assay Kit (Sigma-Aldrich, Milan, Italy), according to the manufacturer’s instructions. Briefly, VSMCs were washed twice with PBS without calcium and magnesium and lysed by HCl 0.6 M treatment for 1 h at 4 °C and overnight at −20 °C to promote decalcification. The intracellular calcium content was determined colorimetrically by interaction with o-creosolphthalein.

### 4.6. IL-6 Determination

According to the manufacturer’s protocol, the IL-6 levels were determined in the culture media using a non-competitive chemiluminescent immunoassay kit (Roche, Mannhein, Germany).

### 4.7. Transmission Electron Microscopy (TEM)

Confluent VSMCs were detached using trypsin and centrifuged at 300× *g* for 5 min. VMSC pellets were fixed in 2.5% glutaraldehyde in 0.1 M cacodylate buffer, pH 7.2, for 2 h at 4 °C and postfixed in 1% osmium tetroxide in the same buffer for 1 h at room temperature. Cells were then dehydrated in a graduated series of ethanol, embedded in Epon-Araldite, and polymerized at 60 °C. Ultrathin sections (60–90 nm), obtained with a Reichert-Jung Ultracut E (Reichert-Yung, Wien, Austria) equipped with a diamond knife, were collected on 200-mesh formvar/carbon-coated copper grids, double stained with aqueous uranyl acetate and lead citrate, and examined with a Jeol 100 SX transmission electron microscope (Jeol, Tokyo, Japan) operating at 80 kV. Micrographs were obtained with an AMTXR80b Camera System (Advanced Microscopy Techniques, Woburn, MA, USA).

### 4.8. RNA Extraction and Real-Time PCR Assay

The total RNA was extracted from mono- and co-culture samples using the RNeasy Plus Micro Kit (Qiagen, S.p.A., Milan, Italy). Briefly, the cells were lysed in an adequate lysis reagent, releasing the nucleic acids in solution and inactivating the endogenous RNase. Genomic DNA was eliminated from the cell lysate (gDNA Eliminator spin column for 30 s at 9000 g), and then the samples were loaded onto micro-columns of silica gel (Rneasy MinElute spin column) to bind the RNA. Finally, the total RNA was eluted in 14 µL of RNAse-free water and stored at −80 °C until used after evaluating integrity, purity, and concentration. The total RNA was reverse-transcribed in first-strand cDNA by miScrpt RTII S Kit (Qiagen). Sirtuin-1 (SIRT-1) and Transforming Growth Factor β1 (TGFβ1) were determined by real-time PCR in the Bio-Rad C1000 TM thermal cycler (CFX-96 Real-Time PCR detection systems, Bio-Rad) and monitored with EvaGreen (SsoFAST EvaGreen Super-mix, Bio-Rad Laboratories, Inc. Hercules, CA, USA). The primers for both control cells and target genes were accurately designed using dedicated software as Beacon Designer ^®^ (version 8.1; Premier Biosoft International, Palo Alto, CA, USA), referring to the nucleotide sequences contained in the GenBank database of the NCBI (http://www.ncbi.nlm.nih.gov/Genbank/index.html, accessed on 1 July 2024) (Table 1). To assess product specificity, amplicons were systematically checked by melting curve analysis. Melting curves were generated from 65 °C to 95 °C with increments of 0.5 °C/cycle. The MIQE Guidelines [36] were followed (Table 2).

### 4.9. Statistical Analysis

Data are expressed as means ± SD for biomolecular analysis, and a *p*-value < 0.05 was considered significant. All data were verified in at least 3 independent experiments and analyzed using GraphPad Prism 8 (GraphPad Software, Boston, MA, USA). The real-time PCR statistical analysis was done using Statview 5.0.1 Software for Windows (SAS Institute, Inc., Cary, NC, USA). The geometric mean of the two most stably expressed genes (eEF1a, RNSP1) and the relative quantification were performed by the ∆∆Ct method using Bio-Rad’s CFX96 manager software version 3.1 (CFX-96 Real-Time PCR detection systems, Bio-Rad Laboratories Inc., Hercules, CA, USA). Group differences were compared using 2-way ANOVA and an appropriate post hoc test for multiple pairwise comparisons. The Pearson correlation test was used to examine the relationship between continuous variables in a linear regression model. Skewed variables were log-transformed before statistical analysis.

## Figures and Tables

**Figure 1 ijms-25-07427-f001:**
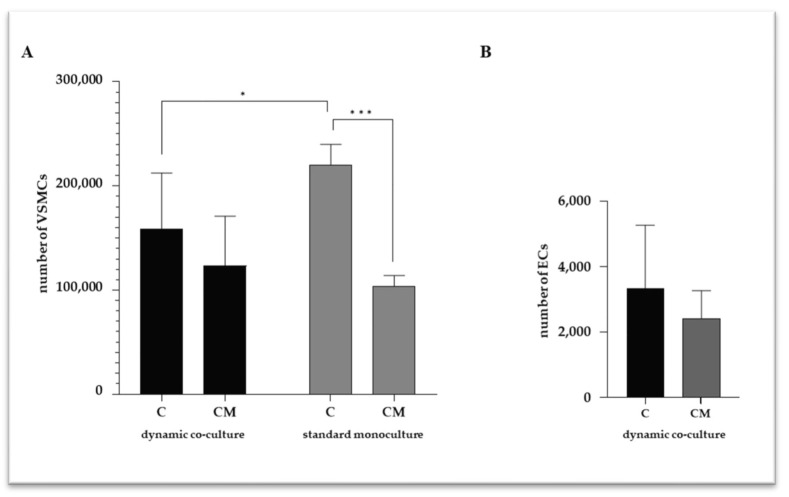
Number of viable VSMCs (**A**) and ECs (**B**). C refers to control cells (VSMCs grown in medium 231); CM refers to VSMCs grown in calcifying medium. Data represent mean of three independent experiments. Statistical analysis was performed with two-way ANOVA and Tukey multiple comparison test; * *p* ≤ 0.05, *** *p* ≤ 0.001.

**Figure 2 ijms-25-07427-f002:**
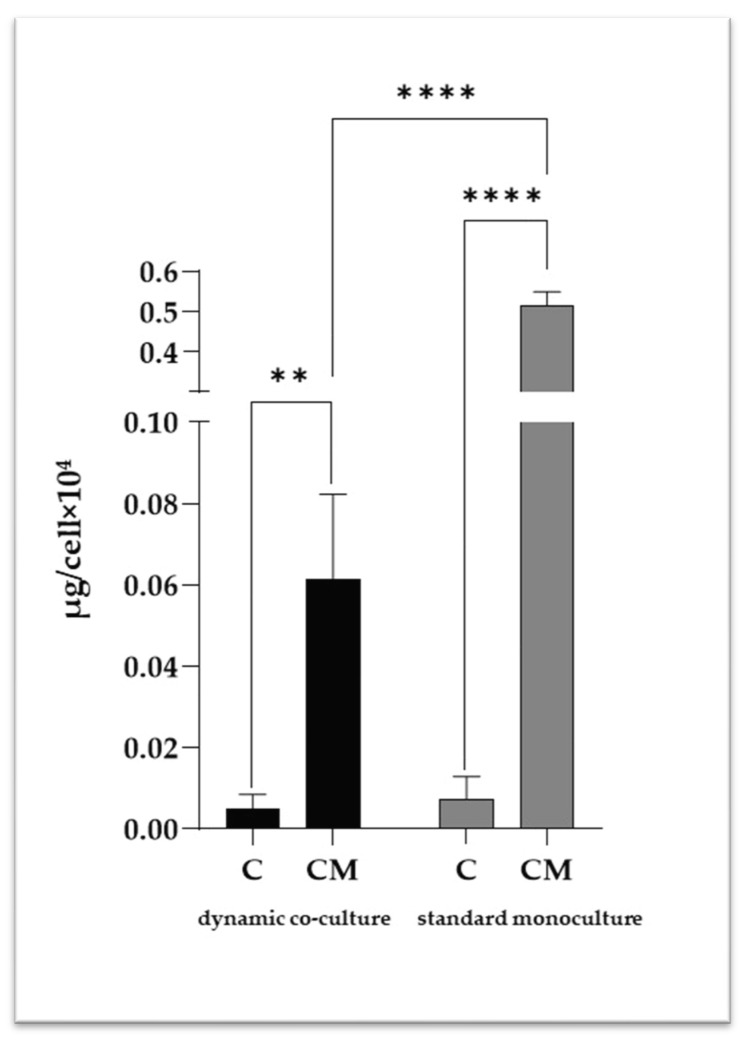
Intracellular calcium quantification in VSMCs. C refers to control cells (VSMCs grown in medium 231); CM refers to VSMCs grown in calcifying medium. Data represent mean of three independent experiments. Statistical analysis was performed with two-way ANOVA and Tukey multiple comparison test; ** *p* ≤ 0.01, **** *p* ≤ 0.0001.

**Figure 3 ijms-25-07427-f003:**
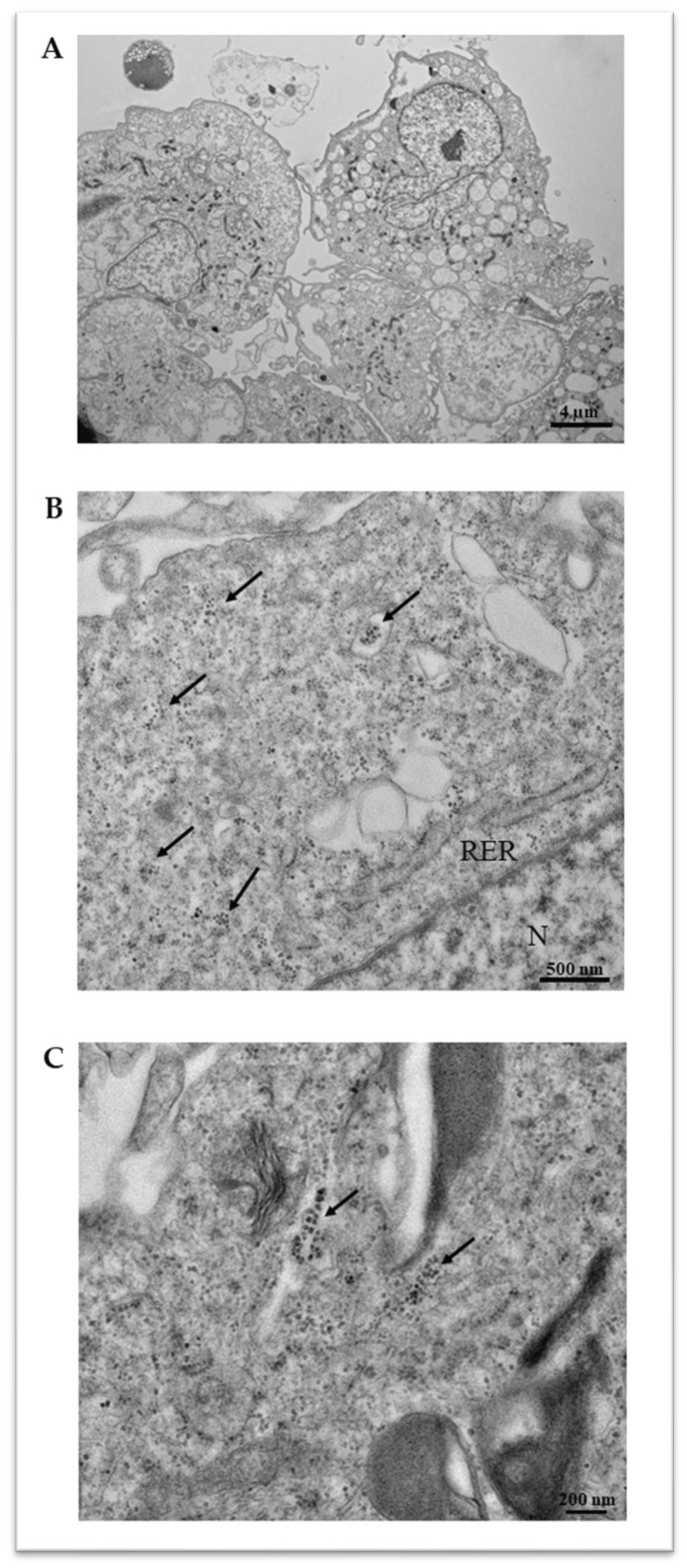
Transmission electron microscopy (TEM) of VSMCs cultured in standard monoculture. Representative images of VSMCs at 1000× (**A**), 8000× (**B**), and 10,000× (**C**) magnification. Intracellular calcium microcrystals (see black arrows) are evident at higher magnification (**B**,**C**), either spread in cytoplasm or inside vesicles. Nucleus (N) and rough endoplasmic reticulum (RER) are indicated. Scale bars equal to 4 µm (**A**), 500 nm (**B**), 200 nm (**C**).

**Figure 4 ijms-25-07427-f004:**
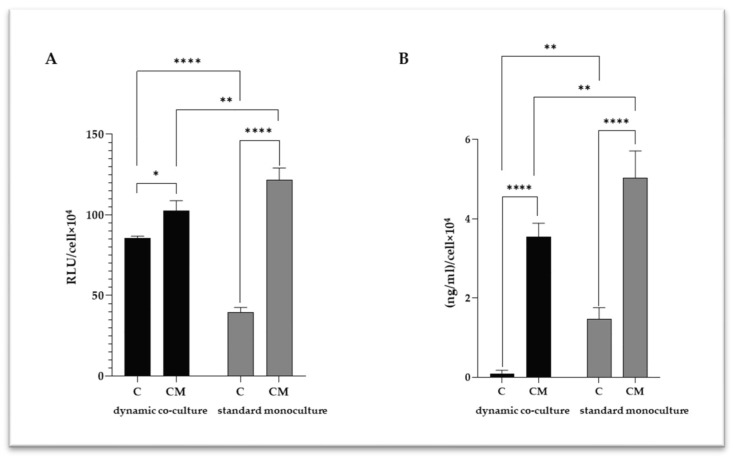
Quantification of caspase-1 (**A**) and IL-6 (**B**) in VSMC culture media. C refers to control cells (VSMC grown in medium 231); CM refers to VSMCs grown in calcifying medium. Data represent mean of three independent experiments. Statistical analysis was performed with two-way ANOVA and Holm–Sidak multiple comparison test; * *p* ≤ 0.05, ** *p* ≤ 0.01, **** *p* ≤ 0.0001.

**Figure 5 ijms-25-07427-f005:**
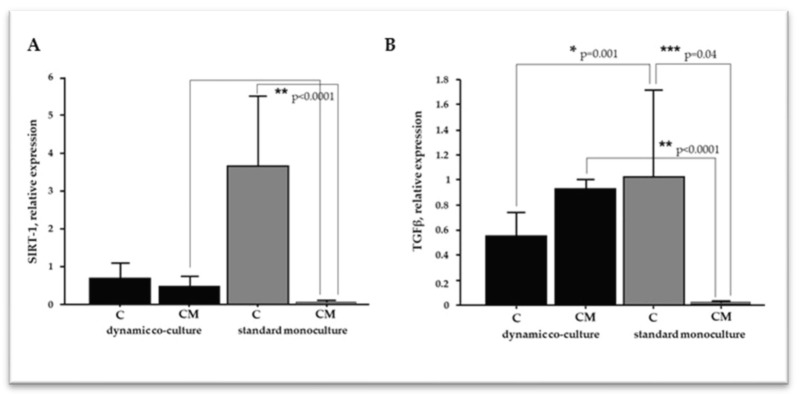
Relative expression of SIRT-1 (**A**) and TGFβ1 (**B**) in VSMCs. C refers to VSMC grown in medium 231 (control cells); CM refers to VSMCs grown in calcifying medium. Data represent mean of three independent experiments, and identified statistical significances were reported.

**Figure 6 ijms-25-07427-f006:**
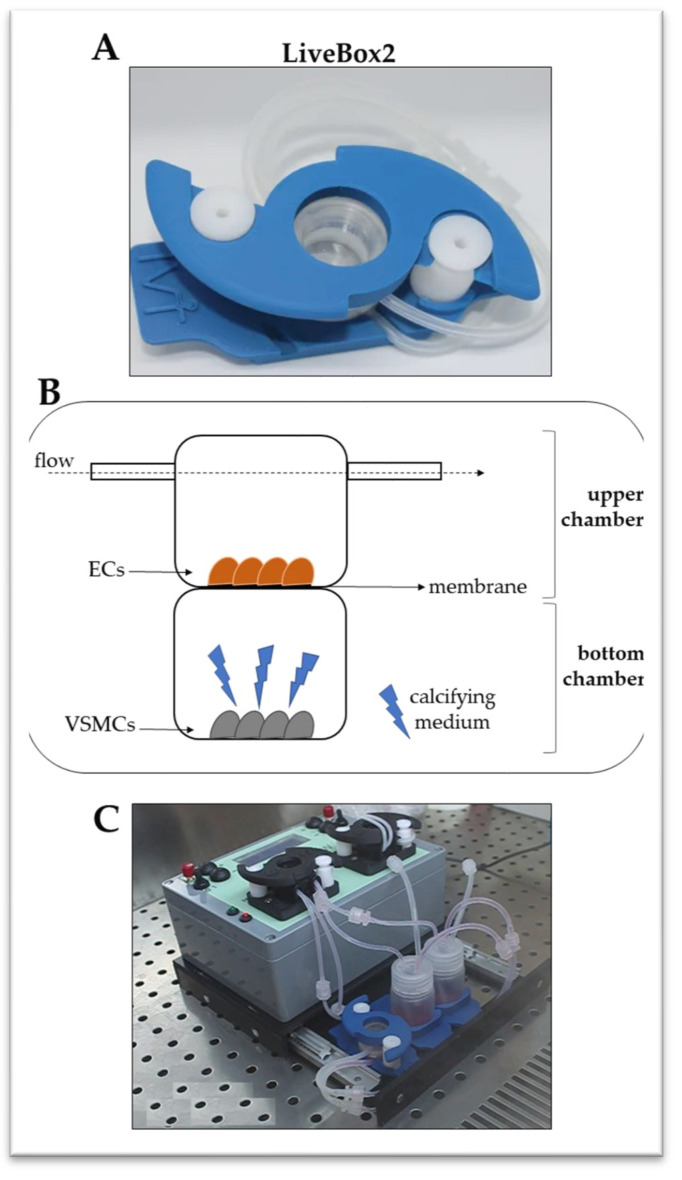
The equipment for the dynamic EC–VSMC co-culture. (**A**). The LB2 cell growth apparatus. (**B**). A representation of the main components of the LB2, with VSMCs grown in the bottom chamber and ECs in the upper one. A permeable membrane separates the two chambers. (**C**). The LB2 connection to the peristaltic pump reproduces the in vivo dynamic conditions.

**Table 1 ijms-25-07427-t001:** Correlation analysis among the intracellular calcium levels and those of caspase-1 and IL-6.

		Caspase-1	IL-6
dynamic co-culture	Intracellular calcium content	R = 0.99 *p* = 0.0155	R = 1.00 *p* = 0.0002
standard monoculture		R = 0.79 *p* = 0.0061	R = 0.81 *p* = 0.0159

**Table 2 ijms-25-07427-t002:** Primer sequences for reference and target genes.

GENE	PRIMER, 5’→3’	Genbank	pb	LOCALIZATION	Ta
*SIRT-1*	F: TCCTCTAGTTCTTGTGGCAGTA R: CATCTCCATCAGTCCCAAATCC	NM_012238	169	chr 10q21.3	58
*TGFβ1*	F: TGAACCCGTGTTGCTCTC R: GCCAGGAATTGTTGCTGTATT	NM_000660.7	104	chr 19q13.2	60

## Data Availability

The original contributions presented in the study are included in the article; further inquiries can be directed to the corresponding author.

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
