# Peer review of "A Dynamic Cellular Model as an Emerging Platform to Reproduce the Complexity of Human Vascular Calcification In Vitro"

_ijms, 2024, doi:10.3390/ijms25137427_

Round 1

Reviewer 1 Report

Comments and Suggestions for Authors

In this manuscript, the authors adopted an EC/SMC co-culture system to investigate the impact of EC and flow on SMC calcification. The concept of co-culture model is certainly interesting to provide a more physiologically relevant VC model. However, there are some major concerns to be addressed in the data and data analysis before the manuscript is acceptable for publication.

1.      In Fig 1, the authors claim the reduction of SMC viability is much less in the dynamic co-culture. However, this is obviously due to the low viability in the control SMC of the co-culture model. The underlying cause is not discussed.

Fig 1A, Fig2 and Table 1 suggests a strong correlation between cell death and calcification. The co-culture model has nearly no cell death and 10-fold lower calcification versus monoculture. The question is whether there is substantial calcification at all in the co-culture model? If the calcification level is too low in the co-culture model, it could disqualify the co-culture model as a calcification model. The authors better employ other calcium quantification methods that give absolute calcium value (e.g. in µg/mg protein), so that the data can be directly compared with other literatures. Also, Figure 3 and relevant text did not specify whether the SEM image of intracellular calcification was observed only in monoculture or in both models. If in both models, SEM images from both models should be presented.

2.      Still in Fig1, SMC counts in all groups are 10,000-20,000, much less than the original seeding number of 30,000. More strikingly, the EC number are <4,000 in all groups, whilst the original seeding number was 45,000. The >90% loss of EC in the co-culture model is not explained, and its impact on SMC behaviour is unclear. TUNEL staining or MTT must be performed to reveal a clearer picture of the cell viability.  

3.      In Fig5, the control levels of SIRT-1 and TGF-beat showed significant difference between the models, and calcification stimulation induced opposite response between the two models. It is difficult to draw a convincing conclusion merely based on the data in Figure 5. More experiments are needed to clarify the roles of these factors.

Author Response

  1. In Fig 1, the authors claim the reduction of SMC viability is much less in the dynamic co-culture. However, this is obviously due to the low viability in the control SMC of the co-culture model. The underlying cause is not discussed.

Given your comments, we have better specified the concepts expressed between lines 72 and 77.

“…the viability reduction for VSMCs cultured in calcifying medium was less pronounced in the dynamic platform, with a viability of about 78% compared to the control cells, whereas in standard monoculture, the viability was about 47%. This phenomenon may be due to the presence of the endothelium under stimuli of flow, conditions that are not present in standard monocultures.”  

Indeed, the viability reduction of calcified VSMC is more mitigated in the dynamic co-culture than in the static one when compared to its own control cells. As you can see in the revised manuscript, we added the percentage of live cells in the calcifying medium compared with the control, which showed that, in the dynamic system, calcified VSMCs are more viable than in the standard monoculture.

2. Fig 1A, Fig2 and Table 1 suggests a strong correlation between cell death and calcification. The co-culture model has nearly no cell death and 10-fold lower calcification versus monoculture. The question is whether there is substantial calcification at all in the co-culture model? If the calcification level is too low in the co-culture model, it could disqualify the co-culture model as a calcification model. The authors better employ other calcium quantification methods that give absolute calcium value (e.g. in µg/mg protein), so that the data can be directly compared with other literatures.

Thank you for your comment. It is true that in our dynamic model, the calcification is lower than in the static monoculture. Still, the key point is that our model more closely resembles the reality of human vascular tissue in terms of cross-relation between endothelial and smooth muscle cells and the presence of the hemodynamic environment. Starting from these considerations, the calcification in monoculture is less realistic. We consider your valuable suggestion about calcium quantification, and in future experiments, we will include it as you suggested.

3. Also, Figure 3 and relevant text did not specify whether the SEM image of intracellular calcification was observed only in monoculture or in both models. If in both models, SEM images from both models should be presented.

TEM is a very accurate methodology that, on the other hand, requires a large number of cells that cannot be obtained from the dynamic model. For this reason, we investigated VSMC calcification using a standard monoculture in T25 culture flasks, using the same calcification protocol in the manuscript. We have revised the caption of Figure 3 to clarify this aspect, and the contrast of Figure 3A has been improved.

4. Still in Fig1, SMC counts in all groups are 10,000-20,000, much less than the original seeding number of 30,000. More strikingly, the EC number are <4,000 in all groups, whilst the original seeding number was 45,000. The >90% loss of EC in the co-culture model is not explained, and its impact on SMC behaviour is unclear. TUNEL staining or MTT must be performed to reveal a clearer picture of the cell viability.  

Endothelial cells were seeded on a semipermeable PET membrane; thus, the adhesion and proliferative capacity decreased with time and due to the presence of calcifying media in the bottom chamber and dynamic flow. The number of viable cells was obtained by measuring viability (by the CellTiter assay described in §4.4) and interpolating this value with a purpose-constructed cell number calibration curve. In accordance with your suggestion, we included this information in section 4.4 of cell viability. However, their presence is sufficient to ensure different effector modulation in VSMCs, as shown in Figure 5, and in the next works, our goal is to optimize the cell adhesion protocol.

5. In Fig5, the control levels of SIRT-1 and TGF-beat showed significant difference between the models, and calcification stimulation induced opposite response between the two models. It is difficult to draw a convincing conclusion merely based on the data in Figure 5. More experiments are needed to clarify the roles of these factors.

Scientific evidence has recognized TGFβ and SIRT-1 as factors related to the change in the VSMC phenotype. The expression trend, identified in the dynamic model, indicates a lower calcifying capacity of VSMCs compared to static monoculture, in which down-regulation affects both effectors. These observations also agree with calcium quantification analysis, which showed higher VSMC calcification in standard monoculture. Although the data are not supported by the existing literature, due to the innovativeness of the model, they are in agreement as a whole. The efforts for the platform characterization are numerous, and we have just submitted a manuscript to the journal Acta Biomaterialia concerning the modulation of Bone Morphogenic Proteins in our VC in vitro model.

Reviewer 2 Report

Comments and Suggestions for Authors

Summary: Authors analyzed intracellular calcium content, inflammation and differentiation of vascular smooth muscle cells (VSMC) using methods of immunometric and molecular biology including comparison of results with those obtained by standard VSMC monoculture. The main aim was to test advanced technology in exploring the pathophysiologic mechanisms underlying vascular calcification. In summary, authors offer experimental model/methodology more suited for revealing the effect of vascular calcification.

Major comment(s)/suggestions:

1.       The whole article should be focused more on methodology the main message supposedly start at rr. 153 and 176 and these are to be at the beginning of the Discussion. The whole article should be more focused on methodology and less on pathophysiology. The exception is that calcification is offered also as the process stabilizing atherosclerotic process and in this respect vessel micro- and macrocalcifications should be discussed (… While macrocalcification confers plaque stability, microcalcification is a key feature of high-risk atheroma and is associated with increased morbidity and mortality…  Irkle, A., et al. Identifying active vascular microcalcification by 18F-sodium fluoride positron emission tomography. Nat Commun 6, 7495 (2015). https://doi.org/10.1038/ncomms8495 , ...).  

2.       Please explain (rr. 63, 78, 220, …) why 7 days of treatment in the calcifying medium were chosen. Different timing could be of essence for such methods.

Minor comments/suggestions:

Some information might be presented in clearer manner:

rr. 54-55 , 153 …“This study investigated the potentiality of VC investigations of a novel dynamic model in which VSMCs and ECs were used as relevant cell types associated with VC. A cell composition that reflects the architecture of the vessel wall applying a mathematical approach to downsize the real human tissue to the in vitro setting was exploited.”

rr. 219-200: … “The LB2 was incubated at 37 °C, 5% CO2 in a humidified cell culture incubator for 7 days, replacing the media following(after?) 72 hours.”

Conclusion: The main strength of this study is well described methodology and rather novel focus on methods how to explore association of inflammatory processes with vascular calcification (in vitro), but some information should be added/discussed in more detail. Pls see my comments above.

Comments on the Quality of English Language

I am not native speaker, but in general some parts could be presented in clearer manner - some highlighted. 

Author Response

  1. The whole article should be focused more on methodology the main message supposedly start at rr. 153 and 176 and these are to be at the beginning of the Discussion. The whole article should be more focused on methodology and less on pathophysiology. The exception is that calcification is offered also as the process stabilizing atherosclerotic process and in this respect vessel micro- and macrocalcifications should be discussed (… While macrocalcification confers plaque stability, microcalcification is a key feature of high-risk atheroma and is associated with increased morbidity and mortality…  Irkle, A., et al.Identifying active vascular microcalcification by 18F-sodium fluoride positron emission tomography. Nat Commun 6, 7495 (2015). https://doi.org/10.1038/ncomms8495 , ...).  

Thank you for your valuable comments. According to your suggestion, we added some information regarding vessel micro- and macrocalcifications into the Discussion section (lines 172-176) and relevant references to support these concepts.

2. Please explain (rr. 63, 78, 220, …) why 7 days of treatment in the calcifying medium were chosen. Different timing could be of essence for such methods.

7 days of treatment was chosen by reviewing relevant articles in the field, which suggest 7 days as sufficient time to induce the presence of calcifications (see reference 26).

3. 54-55 , 153 …“This study investigated the potentiality of VC investigations of a novel dynamic model in which VSMCs and ECs were used as relevant cell types associated with VC. A cell composition that reflects the architecture of the vessel wall applying a mathematical approach to downsize the real human tissue to the in vitro setting was exploited.”

According to your suggestion, we added relevant information regarding the mathematical approach recently published by our research group and included it in the manuscript (see reference 14).“It exploited a mathematical approach to downsize the real human tissue to the in vitro setting, using a cell composition that reflects the architecture of the vessel wall [14]. To this end, this approach considers relevant parameters such as the myocardium mass, the length and the volume of the human coronary tree, the dimension of the coronary endothelium and the smooth muscle layer. According to the mathematical model, the ratio between the ECs and the VSMCs that have to be used in the in vitro model was defined.”

4. 219-200: … “The LB2 was incubated at 37 °C, 5% CO2 in a humidified cell culture incubator for 7 days, replacing the media following(after?) 72 hours.”

Based on your observation, we revised the manuscript where necessary.

5. Conclusion: The main strength of this study is well described methodology and rather novel focus on methods how to explore association of inflammatory processes with vascular calcification (in vitro), but some information should be added/discussed in more detail. Pls see my comments above.

Thank you for your valuable comments. We have revised the manuscript based on your comments and believe we have improved its quality considerably.

Round 2

Reviewer 1 Report

Comments and Suggestions for Authors

I don't think the revised manuscript and the authors' response have addressed any of my major concerns.

Author Response

- I don't think the revised manuscript and the authors' response have addressed any of my major concerns.

We have considered your observations and revised the discussion section to explain better how the cross-talk between endothelial and smooth muscle cells affects the modulation of VSMC behaviour, thus explaining the difference in the results obtained from the two investigated models. Relevant literature has been included to support our data and conclusions. Moreover, we revised the conclusions critically, discussed our results, and pinpointed the limitations and strengths of our model application in vascular calcification studies. We hope the manuscript can address the reviewer's concerns in its present form.